# A Behavioural-Theory-Based Qualitative Study of the Beliefs and Perceptions of Marginalised Populations towards Community Volunteering to Increase Measles Immunisation Coverage in Sabah, Malaysia

**DOI:** 10.3390/vaccines11061056

**Published:** 2023-06-02

**Authors:** Hazeqa Salleh, Richard Avoi, Haryati Abdul Karim, Suhaila Osman, Prabakaran Dhanaraj, Mohd Ali ‘Imran Ab Rahman

**Affiliations:** 1Department of Public Health Medicine, Faculty of Medicine and Health Sciences, University Malaysia, Kota Kinabalu 88400, Sabah, Malaysia; zeqa89@yahoo.com; 2Communications Programme, Faculty of Social Sciences and Humanities, University Malaysia Sabah, Kota Kinabalu 88400, Sabah, Malaysia; haryati@ums.edu.my; 3Sabah State Health Department, Ministry of Health, Kota Kinabalu 88590, Sabah, Malaysia; drsuhaila72@gmail.com; 4Kota Kinabalu District Health Office, Ministry of Health, Kota Kinabalu 88300, Sabah, Malaysia; praba@moh.gov.my; 5Social Preventive Medicine, Faculty of Medicine, University Malaya, Kuala Lumpur 50603, Selangor, Malaysia; aliimran@ummc.edu.my

**Keywords:** health belief model, community volunteering, marginalised populations’ beliefs, measles immunisation coverage

## Abstract

The development of the measles-containing vaccine (MCV) has rendered measles a largely preventable disease. In the state of Sabah in Malaysia, a complete course of measles immunisation for infants involves vaccinations at the ages of six, nine, and twelve months. However, it is difficult for marginalised populations to receive a complete course of measles immunisation. This present study used behavioural theory (BT) to examine the beliefs and perceptions of a marginalised population towards community volunteering as a method of increasing the immunisation coverage of measles. Marginalised populations living in Kota Kinabalu, Sabah, more specifically, Malaysian citizens living in urban slums and squatter areas, as well as legal and illegal migrants, were extensively interviewed in person for this qualitative study. The 40 respondents were either the parents or primary caregivers of at least one child under the age of five. The components of the Health Belief Model were then used to examine the collected data. The respondents had poor awareness of the measles disease and perceived the disease as not severe, with some even refusing immunisation. The perceived barriers to receiving vaccinations included a nomadic lifestyle; issues with finances, citizenship status, language, and weather; failing to remember immunisation schedules; a fear of health care personnel; having too many children; and a lack of female autonomy in vaccine decision-making. However, the respondents were receptive towards community-based programmes and many welcomed a recall or reminder system, especially when the volunteers were family members or neighbours who spoke the same language and knew their village well. A few, however, found it awkward to have volunteers assisting them. Evidence-based decision making may increase measles immunisation coverage in marginalised populations. The components of the Health Belief Model validated that the respondents lacked awareness of the measles disease and viewed it and its effects as not severe. Therefore, future volunteer programmes should prioritise increasing the receptivity and self-control of marginalised populations to overcome barriers that hinder community involvement. A community-based volunteer programme is highly recommended to increase measles immunisation coverage.

## 1. Introduction

“Volunteering” is “any action in which time is spent doing anything that intends to contribute to the environment or people.” [1]. The public health domain has relied on community volunteering strategies to increase immunisation rates by implementing: (i) outreach initiatives, (ii) reminder and recall systems, (iii) education, and (iv) incentives [2]. Community volunteering grew to prominence in Malaysia with the creation of the Malaysia Vaccine Support Volunteers (MyVac) during the COVID-19 pandemic. The MyVac initiative is a nationwide volunteer mobilisation programme that engages individuals who are willing to aid the country’s COVID-19 immunisation programme by reaching out to normally inaccessible groups of people [3]. Community volunteering also significantly helped supplementary immunisation activities (SIAs) for the polio disease in Sabah [4]. Some of the benefits of volunteering include community empowerment; increased resilience and social connectedness [5]; fostering a sense of communal trust, belonging [6], and support; and improving mental health [7].

As such, other health initiatives, including the measles immunisation programme, should adopt community volunteering initiatives more frequently, as measles remains endemic in Malaysia and the goalpost for its eradication has been postponed to 2030. As is the case in many other countries across the globe, the challenges of increasing measles immunisation coverage include low vaccination coverage and vaccine hesitancy in certain demographics, which leaves pockets of individuals susceptible to measles and increases the risk of outbreaks [8]. In Malaysia, measles outbreaks have been reported among nomadic populations in multiple states, owing to their lifestyle [9,10]. Furthermore, there are significant obstacles—such as linguistic, cultural, and geographical restrictions—that prevent undocumented migrants from obtaining immunisation health services [11]. The COVID-19 pandemic caused both positive and negative changes in measles incidences and immunisation coverage in the country. More specifically, in one Malaysian state, the number of measles cases reported in 2019 was higher than 2018 but decreased in 2020 [12]. This decrease could be due to the COVID-19 prevention measures that the country implemented or low measles detection due to inefficient surveillance systems, pandemic-related disruptions to the healthcare system, and decreased propensity to seek medical attention on the part of the patients [13]. Between 2000 and 2021, the MCV is believed to have prevented the deaths of 56 million people worldwide [14,15]. However, in 2021, global measles immunisation coverage was at its lowest since 2008, as only 81% of children received the first dose of the measles-containing vaccine (MCV), thereby, leaving 25 million children susceptible to the disease [14,15].

In Sabah, infants aged six months are given a single dose of the MCV, as they have a higher likelihood of contracting measles than their counterparts in other states. They then receive their measles-mumps-rubella (MMR) vaccinations at nine and twelve months of age, as recommended by the national immunisation schedule [16]. Measles vaccinations were introduced in Malaysia in the early 1950s, under the National Immunisation Programme (NIP), which provides Malaysian citizens with free vaccinations at government-run health centres and hospitals. At present, Malaysia’s measles prevention and control strategies include routine vaccinations, enhanced surveillance, standardised laboratory confirmation, effective case and outbreak management, and specially trained healthcare professionals [17]. Community-based engagement activities should be included in the current strategy to overcome the challenges faced in the eradication of measles.

However, there is unequal representation among the volunteers themselves, as most volunteers are from majority populations [18], and marginalised populations are underrepresented and overlooked. Furthermore, marginalised communities face many daily obstacles that have to be addressed if they are to participate in community volunteering [19]. Despite this, not many studies have examined the perceptions of marginalised communities towards their involvement in community volunteering to improve health issues. Therefore, this present study explores the beliefs and perceptions of marginalised populations towards community volunteering as a method of increasing measles immunisation coverage in Sabah, Malaysia.

## 2. Method

A qualitative research method, by way of in-depth one-on-one interviews, was conducted to document the experiences, perceptions, and beliefs of individuals from marginalised communities.

### 2.1. Study Setting

This present study was conducted in Kota Kinabalu, Sabah, Malaysia—more specifically, Gaya Island (*Pulau Gaya*) and villages under the state’s placement schemes (*Skim Penempatan*)—as most of the state’s marginalised populations live in these locations and measles cases are frequently reported in its unvaccinated migrant communities [20,21,22].

### 2.2. Sample Selection and Recruitment

The respondents of this present study were Malaysian citizens from marginalised communities, living in urban slums and squatter areas, as well as both legal and illegal migrants. Purposeful sampling was used to recruit the intended respondents; namely, parents or caregivers with knowledge of childhood immunisation. Purposeful sampling was used as it guaranteed the inclusion of respondents who possess a wealth of information [23]. Data saturation, the point at which respondents begin providing similar responses, occurred once 40 respondents had been interviewed. As such, no additional data were collected from the subsequent respondents [24]. Extant studies indicate that a cross-cultural or multi-sited study requires a sample size of 20 to 40 respondents to achieve data saturation [25]. Furthermore, a larger sample size may not necessarily yield more valuable data in such studies. Therefore, the sample size of this present study (40) was optimal.

### 2.3. Survey Instrument

A dual-language, semi-structured questionnaire was designed in both English and Malay on the six components of the Health Belief Model (HBM), namely, perceived susceptibility, perceived severity, perceived effectiveness, perceived barriers, cues to action, and self-efficacy, to standardise the process that was used to collect data from the respondents. Nevertheless, the survey instrument did not affect the interview section of this present study, as the answers that the respondents provided were often followed by additional questions from the interviewers. The HBM-based questionnaire was divided into two sections. The first section collected respondent demographics, such as age, gender, occupation, education level, and the immunisation history of their children, while the second examined their beliefs and perceptions of measles immunisation and community-based volunteering. The 10 questions in the second section were open-ended to enable the respondents to share their ideas and opinions on the matter freely. The guidelines of the interview are included in the Appendix A.

### 2.4. Study Procedures

This present study was conducted over a nine-month period—from April to December 2022. The required data were collected by conducting in-depth, one-on-one interviews with the respondents at their residences. The study’s principal investigator, H.S., visited the research site every week and used a semi-structured questionnaire as a guide to conduct the interviews. The mean duration of each interview was approximately 20 to 30 min. The audio of every interview was recorded then transcribed verbatim on a weekly basis. Debriefings were held every week after each phase of data collection to optimise and improve the research method [26]. The interview transcripts and codes were thoroughly reviewed by all members of the research team. Disagreements were discussed and resolved together. As all the respondents provided clear answers, clarification was not required. Throughout the interview period, the number of guided questions remained consistent and none of the questions were deemed unnecessary as all respondents demonstrated a comprehensive understanding of the questions posed. This was because the questions in the Malay language questionnaire were revised to better align with the local dialect. 

### 2.5. Data Analysis

A thematic analysis was conducted using the components of the HBM to theoretically code the collected data [27,28]. According to the HBM, individuals are more or less motivated to perform an action based on their perceptions of their own vulnerability to a problem, the severity of the problem, and the benefits and cost of addressing the problem. Apart from that, internal and external cues to act—from the media, for example—as well as an individual’s confidence in their own ability to act or self-efficacy affect the likelihood of changing a behaviour [29].

Themes were identified from the analysis. ATLAS.ti 22 was used to qualitatively assess the interview transcripts, while two researchers, H.S. and M.A.’I.A.R., coded and analysed them independently. The accuracy of the themes was determined by comparing the obtained results. Conflicting themes were discussed among themselves, as well as with every other member of the research team, to arrive at a final decision. Table 1 depicts how each theoretical construct was used in the present study.

### 2.6. Ethical Approval

Ethical approval for this present study was obtained from the Medical Research and Ethics Committee Ministry of Health, Malaysia, as well as the ethical committee of University Malaysia Sabah prior to the commencement of the study. The confidentiality of the respondents was ensured by not collecting any personal information during the data collection process and anonymising all the collected data. Furthermore, the audio recordings of the interviews were deleted once they had been transcribed.

## 3. Results

In-depth, one-on-one interviews were conducted with the parents or caregivers of children under the age of five from marginalised populations and living in villages on Gaya Island or in villages at the state’s placement schemes. Table 2 provides a summary of the demographic characteristics of the respondents. All 40 respondents were women who were full-time housewives and aged 17 to 64; more specifically, 13 were young mothers aged 17 to 24 years, 26 were aged 25 to 44, and only one was aged 64. Thirteen of the respondents had no formal education while 17 had completed primary school and 10 had completed secondary school. Lastly, four of the respondents had chosen not to vaccinate their children, 30 had had their children vaccinated, two did not complete their children’s measles vaccinations, and the children of four of the respondents had yet to reach the minimum age for the MCV.

### 3.1. Components of the Health Belief Model (HBM)

Direct quotes from the responses of multiple respondents as well as a discussion of the findings of this present study were included. The respondents were anonymised by replacing their names with numbers from 1 to 40 (ID 1 to 40) and classifying the findings. An analysis of the interviews revealed the two main themes: the beliefs and perceptions of the respondents that need to be addressed and promoted according to the components of the HBM. As seen in Figure 1, there were six sub-themes. The terms “few”, “several”, and “most” were used if less than 5, 5 to 10, and 11 to 20 respondents subscribed to a theme, respectively.

#### 3.1.1. Perceived Susceptibility

Only a few respondents lacked knowledge of the measles disease while most believed that children with measles presented with a skin rash. They also believed that their children could contract measles from other children living in high-risk areas. However, a few respondents believed that they were not likely to contract measles as no one in their family had had it. Nevertheless, these same individuals acknowledged that children are susceptible to measles and should, therefore, be vaccinated, as the MCV provides immunity and protection to children. Unfortunately, a few respondents had negative perceptions of the MCV due to the COVID-19 pandemic. Therefore, community volunteers are needed to assist and educate them on the matter.


*“I am not sure. I have never heard of measles.”*

*(ID 2)*



*“I have heard of measles. Children exhibit skin rashes.”*

*(ID 30).*



*“We are susceptible to measles as it could be anywhere. Some children in this village have skin rashes which could, possibly, be measles.”*

*(ID 25).*



*“None of my family members have ever had measles. So, we probably have low susceptibility to measles.”*

*(ID 22).*



*“At present, as all my family members are completely vaccinated so, we are protected from measles. It is dangerous for children if they are not vaccinated as they have a higher likelihood of contracting the disease.”*

*(ID 1).*



*“I have heard of measles. The MCV increases the number of antibodies required to fight the disease.”*

*(ID 10).*



*“Since the beginning of the COVID-19 pandemic, many people have refused to take their children to clinics to get vaccinated. Therefore, someone should come and explain so that people feel confident about vaccinating their children.”*

*(ID 29).*


#### 3.1.2. Perceived Severity

Most of the respondents viewed contracting measles as a trivial health issue, as children would only develop skin rashes and a fever that could be treated with medicine. Furthermore, only a few respondents believed that the MCV was safe as their children did not develop any complications post-immunisation.


*“It is normal for children to have a fever and skin rashes. If they are eating and drinking like normal, it means they will recover.”*

*(ID 39).*



*“My children usually develop a fever post-vaccination but they recover after taking anti-pyretic medication. So, I am not too worried about the side effects of the MCV.”*

*(ID 1).*



*“This is my third child and his vaccinations are up-to-date. He did not develop any side effects following his measles vaccination.”*

*(ID 5).*


#### 3.1.3. Perceived Benefit

Most of the respondents believe that measles vaccination is needed for the health of their children. They also felt that a community volunteer programme could help get children vaccinated. A few respondents believed that volunteers could help explain, share knowledge, and remind parents of their children’s vaccination appointments. They also felt that volunteers could help encourage, placate, and convince individuals who distrusted childhood vaccinations after the COVID-19 pandemic that their children needed the MCV. Volunteers could also act as spokespersons of village communities and communicate with health care professionals when individuals are unable to visit a clinic in person. However, several respondents felt that it is better and more beneficial to deliver the free MCVs to villages, as they would not have to go to a clinic then.


*“So long as my children are vaccinated, it is okay. It is for their health after all.”*

*(ID 24).*



*“Having a community volunteer may be beneficial as they can help children get their MCVs.”*

*(ID 9).*



*“I am not sure how community volunteers could help. Maybe if they promote immunisation, more people would understand and take their children to be vaccinated.”*

*(ID 20).*



*“As the volunteers can explain immunisation, it would help the people in this village understand better.”*

*(ID 7).*



*“I agree with this programme as it could help remind me of my children’s appointment. I sometimes I forget.”*

*(ID 9).*



*“Many people here are afraid of vaccinating their children because of the COVID-19 pandemic. Maybe if the volunteer is from our own village and encourages vaccination, more people will be confident about vaccinating their children.”*

*(ID 28).*



*“If there are community volunteers in this village, maybe the people of this village will be less afraid of asking questions about vaccination.”*

*(ID 8).*



*“It will be good if the community volunteers can help inform the nurses when we are unable to attend a vaccination appointment at the clinic as it is difficult to travel when the area is flooded.”*

*(ID 37).*



*“I agree with the community volunteer programme. If there are people coming to the village to vaccinate the children, they can educate other villagers and neighbouring villages so they can all bring their children with them.”*

*(ID 27).*



*“The community volunteer programme seems okay to me. It would be even better if people can come and administer the free MCV to the children. After all, it is for their health.”*

*(ID 36).*


#### 3.1.4. Perceived Barriers

Nevertheless, parents and caregivers from marginalised populations face multiple obstacles such as those related to finances, language, legal issues, family, and bad weather. Furthermore, many have to travel long distances to the clinic on foot, which is tiring. A few respondents said that they were afraid of asking the nurses questions about immunisation, as they are often scolded, even if they ask politely. Some also admit that they forget immunisation appointments and do not receive reminders or messages from the nurses. Meanwhile, a few believed that their nomadic lifestyle makes it difficult to attend follow-up appointments. Furthermore, in some of these marginalised families, childhood vaccination decisions are made by the father, who is also the sole breadwinner. Therefore, if the father disagrees, the mother has no choice but to comply. Some of the respondents are also anti-vaccine and refuse to vaccinate their children. Although most of the respondents believed that a community volunteer programme is beneficial and would help overcome the aforementioned barriers, some were uncomfortable dealing with strangers.


*“What is the point of taking my children to a clinic to be vaccinated if I do not have money?”*

*(ID 11).*



*“I cannot vaccinate my children at a clinic because I do not have legal documentation. Even if I did have the required documents, I cannot travel there as I cannot afford it.”*

*(ID 36).*



*“The villagers need to help each other because many do not understand or speak Malay. Most of them only speak Suluk amongst themselves.”*

*(ID 32).*



*“I have many children, so if there is nobody at home to look after them, I have to bring all of them along with me to the clinic. It is very difficult for me to bring all of them along.”*

*(ID 21).*



*“The area sometimes floods when it rains heavily. So, it is difficult to travel to the clinic when it is flooded.”*

*(ID 25).*



*“It is difficult to travel to the clinic when the sea is rough. When it is too windy, I cancel my plans to go to the clinic.”*

*(ID 13).*



*“I have no problem travelling to the clinic but the walk is quite tiring as I am pregnant. On the island, I have to walk to the boat at the town’s jetty. Then I have to walk further to reach the clinic. It takes about a half an hour of walking from the island to reach the clinic.”*

*(ID 9).*



*“If I had an illness that I was unfamiliar with, I would ask my family or the nurses at the clinic. But I would prefer to ask my family first because I am scared that the nurses will scold me.”*

*(ID 27).*



*“I usually ask the nurses but I always get scolded. They always answer me as if they are angry. Even when I ask them politely, I still get scolded.”*

*(ID 16).*



*“I often forget the children’s vaccination appointments and the nurses never remind me.”*

*(ID 1).*



*“I do not have any problem taking my children to the clinic to be vaccinated. I used to go to the Maternal and Child Health Clinic (Klinik Kesihatan Ibu dan Anak, KKIA) in Pekan. Now that I have I moved here with my husband, I go to a clinic in Telipok, which is nearer and only five minutes travel by taxi. But some of my children missed their vaccination appointments because I moved around a lot.”*

*(ID 28).*



*“I always ask my husband before doing anything. If he does not agree, then there is nothing that I can do about it.”*

*(ID 30).*



*“Some of my nephews and nieces back home are unvaccinated. Some of my family members are anti-vaccine.”*

*(ID 8).*



*“Even if the community volunteers are from this village, I would not be confident about vaccinations. Maybe because I do not know them well enough.”*

*(ID 19).*


#### 3.1.5. Cues to Action

Most of the respondents were receptive towards community volunteer programmes and believed it was a good strategy as the volunteers would be from their own village, which mostly consists of family members or acquaintances. The respondents also believed that it would be more beneficial if the volunteers spoke their language. Another cue to action was volunteers who are more familiar with the surroundings that they live in.


*“I would not mind if there was a community volunteer at the village. Most of the villagers are my relatives anyway.”*

*(ID 3).*



*“Some of the villagers do not understand Malay. So, a volunteer who can actually help would really help the situation.”*

*(ID 30).*



*“I do not mind having a community volunteer in the village even though I do not understand how it would help. In any case, the villagers often help nurses locate patients in the village.”*

*(ID 17).*


#### 3.1.6. Self-Efficacy

Most respondents believed that it was important to vaccinate their children. As such, they made an effort to have their children vaccinated. Meanwhile, a few turned to the nurses or their family members if they had immunisation-related questions.


*“All the children that I look after are fully vaccinated. Now that I am also taking care of my grandchildren, I made sure that they were fully vaccinated as their parents work far away in town.”*

*(ID 34).*



*“If I do not know something, I will ask the nurses. They always answer all my questions.”*

*(ID 18).*



*“I asked my family members and my husband. They do not have any qualms about vaccinating the children.”*

*(ID 2)*


## 4. Discussions

### 4.1. Key Findings

All the respondents of this present study were from marginalised populations. Most of them were completely unaware of measles while a few were unaware of both the disease and the importance of immunisation. Even the more educated respondents had little knowledge of measles. Extant studies on migrant communities in other nations have, similarly, reported that more than half had little to no understanding of measles, the MCV, and the immunisation schedule [30,31]. However, individuals with more information about the MCV were more confident and exhibited higher vaccine uptake [32]. A correlation has also been discovered between the education level of parents and caregivers and vaccine uptake. More specifically, vaccine uptake is higher among individuals with a secondary school education or higher than their less educated or uneducated counterparts [33]. Nevertheless, the correlation between vaccine hesitancy and education level is debatable. For instance, in some cases, vaccine hesitancy is higher among well-educated parents and caregivers due to their social status and better access to vaccine-related information [34].

There are opposing views on the subject of measles vaccination in marginalised populations. While some believe that the MCV is safe and effectively immunises children, others disagree as they were sceptical of COVID-19 vaccines and generalised this scepticism to other vaccines, such as the MCV. Individuals with anti-measles antibody levels that exceed 200 mIU/mL are immune, while individuals with lower levels are less likely to infect others [35]. The COVID-19 pandemic also altered the attitudes of mothers towards childhood vaccinations and resulted in decisions to leave their children unvaccinated [36]. An increase in vaccination scepticism often coincides with the release of controversial new data, policies, or reports on the potential dangers of vaccines [37]. This has only increased vaccine hesitancy and scepticism in a population that is already distrustful and avoids using public health services [38]. Marginalised and majority populations in Malaysia have different concerns regarding vaccine hesitancy. Most Malaysian parents express concerns about the safety, side effects, and efficacy of vaccines as well as myths, such as that children are currently receiving too many immunisation shots and that natural infection is a better way of building immunity [39]. Furthermore, religion, superstition, and a preference for holistic medicine further add to the general mistrust of conventional medicine and healthcare providers [40].

The respondents of this present study did not view measles as a severe disease that could endanger the lives of their children. As they believed that the MCV would not adversely affect their children, they did not deem it important enough to warrant concern. Therefore, low perception of disease severity and the associated risks, coupled with complacency, leads to lower vaccine uptake. A study that was conducted in the United Kingdom (UK) similarly found that the general public is unaware that measles can be fatal and cause serious complications [41]. Pneumonia, a common complication in unvaccinated children with measles [42], is fatal in children under the age of five.

The barriers to vaccination identified in this present study are similar to that of other studies of marginalised populations in Sabah. More specifically, it is difficult for illegal migrants to vaccinate their children, as they lack funds and proof of legal citizenship as well as face language barriers, COVID-19 concerns, and weather-related impediments to health services [11]. In Malaysia, the most common reason why parents fail to fully vaccinate their children include busy work schedules and forgetting the vaccination appointment [43]. A nomadic lifestyle is also a barrier to increasing immunisation uptake. There is only a handful of policies that protect nomadic populations from the many diseases that pose a far greater threat to them [44]. A nomadic lifestyle also significantly correlates with missed vaccination opportunities, even if a health facility was visited for other health concerns [45]. This present study also found that women lack the autonomy to decide on the vaccination status of their children. An Indonesian study similarly found that one of the biggest hurdles that women faced when trying to vaccinate their children was opposition from their partners [46], while a Ghanaian study found that the unequal position of women in society limits their ability to make healthcare decisions, even for themselves [47].

Nevertheless, most of the respondents of this present study believed that community volunteering would increase the uptake of the MCV. Furthermore, their expectations for this programme indicate that some barriers to immunisation may be overcome once the programme is initiated. After all, telephone reminders effectively improved immunisation coverage in low- and middle-income countries [48]. As a recall and reminder system may directly affect the behaviour of an individual without changing his or her thoughts or feelings on the matter, it an excellent intervention for vaccine uptake [49]. Furthermore, community interventions are also effective at boosting immunisation rates [50].

### 4.2. Programme Framing and Target

Therefore, a community volunteer programme should aim to increase the perceived susceptibility and severity of measles. Perceived susceptibility is a contextual factor as a mother is more likely to vaccinate her child if she acknowledges the possibility of her child contracting the disease. Changing the views of individuals enables them to alter their behaviours on their own [50]. High perceived susceptibility could increase vaccine uptake among older adults [51], while low perceived susceptibility often correlates with vaccine hesitancy among parents and caregivers as they have a low perception of the risks of the disease [52]. Once a parent or caregiver accepts that their children are susceptible to a disease, they take the severity of the illness into consideration [53].

Improving self-efficacy also correlates with the adoption of recommended behaviours. Several studies have found that an adult or parent’s sense of self-efficacy in relation to vaccination are significant predictors of their actual vaccination behaviour [54,55]. This present study found that marginalised populations are more comfortable interacting with their own community. Although they are generally accepting of outsiders, especially those who share the same culture as them and understand their way of life, some are uncomfortable with strangers. However, choosing community volunteers that they are acquainted and familiar with would overcome this issue. Furthermore, obtaining the trust and confidence of the community in the volunteer programme would encourage the community to be more self-sufficient in the long run. Peer pressure has also been found to affect multiple health-related behaviours, including vaccination [56]. Nevertheless, conflicting results have been reported when examining the correlation between societal pressure and immunisation. For instance, exposure to vaccinated individuals was found to reassure those around them of the safety and efficacy of a vaccine and lead to higher immunisation rates. However, achieving herd immunity may discourage individuals from getting vaccinated [57].

Communicating information and attracting attention should be a priority when the programme is initiated. Highlighting the benefits of the community volunteer programme is an option. The media is an effective method of communicating the importance of vaccination and garnering public support. Conversely, poor communication can aggravate vaccine scepticism and lower vaccination rates. The message and language used to outline vaccine goals to a marginalised population must be comprehensible if they are to trust immunisations and increase vaccination rates [58]. Furthermore, highlighting the dangers of vaccination certainly decreases the intentions of individuals to get vaccinated. Therefore, when informing others of vaccination, the dangers should be vaguely expressed while an emphasis should be placed on lowering the risk of infection. Lastly, humour has been found to attract and lower reactance which, in turn, decreases vaccine hesitancy in a community [59].

### 4.3. Strengths and Limitations

This present study used the components of the HBM to evaluate the ability of community volunteering initiatives to improve the completion of measles immunisation and coverage among marginalised populations in Sabah, Malaysia. As this household-based study detects lost opportunities at a family level, it is a better representation of the situation on the ground than studies conducted by healthcare facilities. Nevertheless, the sample of this study was predominantly female, as it is common in marginalised populations for women to care for the children and perform household chores while the men work to support the family. As this present study conducted in-depth, one-one-one interviews, the responses of every respondent were wholly unique and uninfluenced by the others. The present study was conducted in two areas in Kota Kinabalu with high concentrations of marginalised peoples: villages on Gaya Island and villages at Placement Schemes. The 40 respondents of this present study were from villages in these two areas.

This present study is not without its limitations. Firstly, only two sites were studied, due to budgetary and time constraints. Secondly, the findings of this present study may suffer from information bias, as all the respondents were women. Thirdly, a bigger sample size with respondents from other villages would have provided a better representation of marginalised populations in Sabah. Lastly, a pilot study using a semi-structured questionnaire should have been conducted prior to the commencement of the present study, as it may have improved the quality of the study.

## 5. Conclusions

Evidence-based decisions should be used to develop measures to increase the coverage of the MCV in marginalised populations. Multiple studies have proposed interventions and behavioural, theory-based strategies to increase vaccine uptake. However, different populations in different countries respond differently. Therefore, this present study fills a significant gap in the existing literature on measures to increase immunisation coverage. It also provides evidence for future actions and policies in Malaysia, particularly in the state of Sabah. The findings of this present study also provide a better understanding of the prospective messages and message delivery methods that can be used by community volunteer programmes in middle-income countries, especially to garner the participation of marginalised populations. One method is the implementation of a straightforward recall and reminder system that is run by the community in question. This way, the community would encourage, inform, and persuade parents and caregivers to have their children vaccinated. Future studies could examine methods of implementing community volunteer programmes, as well as assess the success and impact of such programmes in increasing the coverage of measles immunisation.

## 6. Contributions to Literature

This present study adds to the body of knowledge by describing the perceived susceptibility and severity that marginalised populations have of measles. It also outlines the benefits and barriers that community volunteering programmes face in increasing the coverage of measles immunisation.

A better understanding of the needs, challenges, cues to action, and self-efficacy of marginalised populations is also elucidated, in order to develop better and more tailored programmes for them.

Lastly, this present study provides an idea of the willingness of marginalised populations to be involved in community volunteering programmes.

## Figures and Tables

**Figure 1 vaccines-11-01056-f001:**
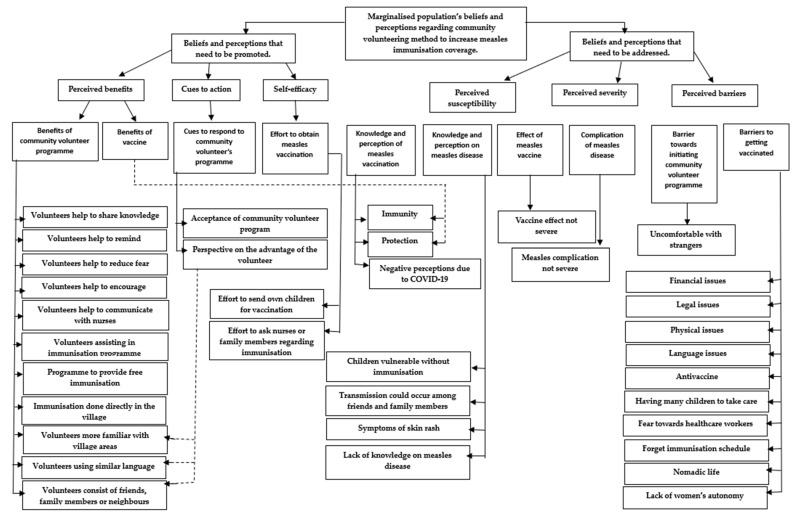
Mapping of the themes.

**Table 1 vaccines-11-01056-t001:** The Health Belief Model Constructs.

Behavioural Theory	Construct	Brief Definition
Health Belief Model	Perceived susceptibility	Beliefs about the likelihood of a child contracting measles.
	Perceived severity	Beliefs about the severity of the consequences of a child contracting measles.
	Perceived benefits	Beliefs about the efficacy of measles vaccination, as well as community volunteers to help reduce the risk and severity of the problem.
	Perceived barriers to action	Beliefs on the financial and emotional costs of receiving the measles vaccine, as well as the need for community volunteers to help with the programme.
	Cues to action	Strategies to increase a person’s willingness to act to lower the chance of a child contracting measles.
	Self-efficacy	The belief that one can take measures and overcome obstacles to prevent the spread of measles among children.

**Table 2 vaccines-11-01056-t002:** Focus Group Participant Characteristics.

Characteristics		*n*	%
Gender	Male	0	0
	Female	40	100
Caregiver Status	Parent	38	95
	Caregiver	2	5
Age Range	17–24 years	13	33
	25–34 years	18	45
	35–44 years	8	20
	45–64 years	1	3
Educational level	High school	10	25
	Primary school	17	43
	No schooling	13	33
Occupation	None	40	100
History of Children Measles Immunisation Status	Complete	30	75
	Incomplete	2	5
	Nil	4	10
	Not yet	4	10
Village	*Pulau Gaya*	21	52
	*Skim Penempatan*	19	48

## Data Availability

The data presented in this study are available on request from the corresponding author.

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
