# Peer review of "A Behavioural-Theory-Based Qualitative Study of the Beliefs and Perceptions of Marginalised Populations towards Community Volunteering to Increase Measles Immunisation Coverage in Sabah, Malaysia"

_vaccines, 2023, doi:10.3390/vaccines11061056_

Round 1

Reviewer 1 Report

The article entitled “A Qualitative Study Using Behavioural Theory to Explore the Marginalised Population’s Belief and Perception Regarding Community Volunteering Method to Increase Measles Immunisation Coverage in Sabah, Malaysia” by H. Salleh et al. reported a study on the behavior of a limited population concerning Measles vaccine in Malaysia. Patients were interviewed (n=40) using a standardized process.

The acceptation of vaccination by a poor under educated population was variable and sometimes limited for different reasons.

The authors proposed a program to improve the access to vaccination which should be adapted to the patient community. According to what has been made in similar countries the authors are convinced that a program should be developed but are uncertain about the potential results. The article described clearly what is the health policy and what could be improved.

The English language should be edited. 

Minor revisions

The figure 1 “Mapping of the themes, should be splitted in two parts:

A)

Perceived susceptibility

Perceived severity

Perceived barriers

B)

Perceived benefits

Cues to action

Self-efficacy

Results 3-1-1 Perceived susceptibility

The significance of the abbreviations should be indicated, for example line 173 (ID2) line 176 (ID30)…

Author Response

Point 1: The article entitled “A Qualitative Study Using Behavioural Theory to Explore the Marginalised Population’s Belief and Perception Regarding Community Volunteering Method to Increase Measles Immunisation Coverage in Sabah, Malaysia” by H. Salleh et al. reported a study on the behavior of a limited population concerning Measles vaccine in Malaysia. Patients were interviewed (n=40) using a standardized process. The acceptation of vaccination by a poor under educated population was variable and sometimes limited for different reasons.The authors proposed a program to improve the access to vaccination which should be adapted to the patient community. According to what has been made in similar countries the authors are convinced that a program should be developed but are uncertain about the potential results. The article described clearly what is the health policy and what could be improved.

The English language should be edited. 

Minor revisions

The figure 1 “Mapping of the themes, should be splitted in two parts:

A)

Perceived susceptibility

Perceived severity

Perceived barriers

B)

Perceived benefits

Cues to action

Self-efficacy

Response 1: Mapping of the themes had been splitted to two parts as recommended by the reviewer. Refer Figure 1- Mapping of Themes. English languange had been proofread prior to journal submission.

Page 10

Point 2: Results 3-1-1 Perceived susceptibility

The significance of the abbreviations should be indicated, for example line 173 (ID2) line 176 (ID30)…

Response 2: Significance of the abbreviations had been added.

Changes to text:

Page 5 ,Line 183-185

Direct quotes from different respondents as well as a discussion of the study’s findings were included. Respondents were re-identified from number one to 40 (ID 1-40) to maintain their confidentiality and classified the findings.

Reviewer 2 Report

This study uses data from semi-structured interviews conducted among residents of Kota Kinabalu, Sabah who had at least one child under the age of five during April-December 2022 to explore beliefs and perceptions about a community volunteer measles immunization programme. The following points will help strengthen the manuscript.

1. Lines 1-5: Please ensure that there are no typos in the Title.
2. Lines 14-37: Please ensure that there are no typos in the Abstract.
3. Lines 42-125: Please consider shortening the Introduction section making it concise. The last paragraph should clearly state the gap in the literature and how this study aims to fill that gap. Also, the last paragraph should not be used to introduce new information such as “behavioural theory” (which can be done later in the Methods section).
4. Line 51: “(MyVac, 2022)”?
5. Line 67: “Western Pacific Region”?
6. Lines 127-129: Please add a Setting subsection here providing information about the choice of “Kota Kinabalu, Sabah” for this study. Later, authors report that interviews were conducted “in the villages of Pulau Gaya and Skim Penempatan Telipok” (lines 159-161) and then, towards the end, they identify “Gaya Island and Skim Penempatan Telipok” as 2 “areas” (lines 435-437). Are these the only 2 villages in “Kota Kinabalu, Sabah”? If not, please provide the rationale for focusing on these 2 villages. Or are these the 2 “areas”… Please be clear and precise.
7. Line 133-134: Please provide a definition of “purposeful sampling”?
8. Line 133-134: How was the sample saturation determined in this study? Please be specific.
9. Lines 134-136: Please explain what is meant by expressions such as “…will be met…” and “participants will then be approached…”?
10. Lines 138-139: Please provide a section to explain what was done to secure the confidentiality and privacy of participant information while they were interviewed in their homes.
11. Lines 138-139: Please be specific about the number of interviews conducted, indicating how “40” was determined. Again, how was the sample saturation determined in this study?
12. Lines 139-142: Please describe the semi-structured interview guide development process. Was this interview guide pilot tested? Please describe this process in detail.
13. Lines 139-142: How many questions were there in this interview guide initially?
14. Lines 139-142: Please provide a concise description of the “Health Belief Model” here and describe how it informed the interview guide development.
15. Lines 139-142: Please describe if there were any debriefings after interviews were conducted?
16. Lines 139-142: Please describe if there were any iterative refinement of the interview guide. Did the number of questions change as a result? Please be specific.
17. Lines 139-142: Please provide a copy of the interview guide in an appendix.
18. Lines 142-145: It is better practice that the initials of the “principal investigator” is provided. Is the “principal investigator” and “primary researcher” the same individual? Please be precise.
19. Lines 147-156: This reviewer is very concerned that the analytical process was not conducted by at least 2 authors independently which brings into question the credibility of results.
20. Lines 147-156 and Table 1: Please describe the data analysis process clearly (for example, is it the use of “content analysis” that “employed parts of the Health Belief Model…to code data…” or is it the “inductive analysis of qualitative data” that “identified common themes…” as in here). Also, see the Title and lines 427-429 for similar concerns that this reviewer have given that the analysis process is not clearly described.
21. Lines 147-156: Again, who is “principal analyst”? Please provide initials.
22. Lines 147-156: Again, which “member of the research team”? Please provide initials.
23. Lines 147-156: What did that process of “check[ing] the coding afterwards” involve?
24. Lines 154-155: How are “groups” similar or different from “themes”?
25. Lines 155-156: How was it possible to “finally find recurring patterns or themes”?
26. After line 156: Please add here to describe if there were ‘member checking’.
27. Lines 159-164 & Table 2: “Nil”?
28. The rest of the sections of this manuscript: Given above points, this reviewer is unsure in terms of how to continue with the rest of the section of this manuscript other than to indicate that direct quotes should be translated to English (since the rest of manuscript is in English).
29. Please avoid typos and ensure completeness, consistency and transparency in the manuscript: a) please make sure that all citations are referenced in a similar format in the body of the manuscript (see, for example, “Anna A. Minta….” in line 68, why is this citation in a different format?), b) please ensure all citations in the body of the manuscript are also reported in the Reference section (for example, see above for the missing one), c) please ensure completeness of citations and follow the same format throughout the Reference section (for example, why are some journal names abbreviated and others not, why some citations start with first name of the author and others with the last name, why are some citations in capital letters, what does “xxxx” mean…), d) please be transparent (which “report, line 68; which “region” line 69…), e) please write out all acronyms (such as, “MCV” in line 78 or “HBM” in line 119) the first time they are mentioned, f) please do not repeat writing out the acronyms later in the manuscript (see lines 166-170 for “HBM”), g)  please ensure consistency (see, for example, is it “majority…” or is it “only a few…” lines 317-319), and h) please proofread before submission (see, for example, “…with other literature in a According to a Steven et al. study…” lines 359-361)…

Author Response

Point 1: This study uses data from semi-structured interviews conducted among residents of Kota Kinabalu, Sabah who had at least one child under the age of five during April-December 2022 to explore beliefs and perceptions about a community volunteer measles immunization programme. The following points will help strengthen the manuscript.

Lines 1-5: Please ensure that there are no typos in the Title.

Response 1: Manuscript was sent for proofreading and editing prior to journal submission. Certificates attached.

Point 2: Lines 14-37: Please ensure that there are no typos in the Abstract.

Response 2: Manuscript was sent for proofreading and editing prior to journal submission. Certificates attached.

Point 3: Lines 42-125: Please consider shortening the Introduction section making it concise. The last paragraph should clearly state the gap in the literature and how this study aims to fill that gap. Also, the last paragraph should not be used to introduce new information such as “behavioural theory” (which can be done later in the Methods section).

Response 3: Introduction section was adjusted to be more concised. Behavioural theory was moved to the method section.

Changes to text:

Introduction: Page 2-3, Line 47-114

Behavioural Theory: Page 4, Line 148-153

The thematic analysis employed parts of the Health Belief Model as a theoretical lens to code data [28,29]. The HBM postulates that people are more or less motivated to take action based on their judgements of their own vulnerability, the severity of a problem, and the benefits and costs of addressing that problem. The likelihood of behaviour change is also affected by things like internal and external cues to act (like the media) and a person’s confidence in their own ability to act (self-efficacy) [30].

Point 4: Line 51: “(MyVac, 2022)”?

Response 4: The reference had been revised.

Changes to text:

Page 2, Line 51-56

In Malaysia, community volunteering became prominent during the COVID-19 pandemic through the creation of Malaysia Vaccine Support Volunteers (MyVac). It was a nation-wide volunteer mobilisation pro-gramme that aimed to engage individuals willing to assist in the COVID-19 immunisation programme to reach an inaccessible group of people [3].

Point 5: Line 67: “Western Pacific Region”?

Response 5: The sentence Western Pacific Region had been changed to worldwide.

Changes to text:

Page 2 Line 76-79

In addition, the estimated worldwide measles immunisation coverage also dropped, with only 81% of children having received their first dose of measles-containing vaccine (MCV) in 2021, resulting in 25 million children being susceptible to measles [14,15].

Point 6: Lines 127-129: Please add a Setting subsection here providing information about the choice of “Kota Kinabalu, Sabah” for this study. Later, authors report that interviews were conducted “in the villages of Pulau Gaya and Skim Penempatan Telipok” (lines 159-161) and then, towards the end, they identify “Gaya Island and Skim Penempatan Telipok” as 2 “areas” (lines 435-437). Are these the only 2 villages in “Kota Kinabalu, Sabah”? If not, please provide the rationale for focusing on these 2 villages. Or are these the 2 “areas”… Please be clear and precise.

Response 6: Study setting subsection is added where the study sites were standardised throughout the manuscript. Rationale was provided.

Changes to text:

Page 3, Line 118-121

This research was conducted in Kota Kinabalu, where instances of measles were frequently reported among migrants who had not received immunisation [21]. The selected locations for the study were Pulau Gaya and Skim Penempatan, which were recognised as primary areas of residence for the marginalised population in Kota Kinabalu [22,23].

Point 7: Line 133-134: Please provide a definition of “purposeful sampling”?

Response 7: Definition of purposeful sampling is added.

Changes to text:

Page 3, Line 124-128

The recruitment process employed a "purposeful sampling" technique, as the intended participants were individuals who serve as parents or carers with knowledge of childhood immunisation. The utilisation of this approach guaranteed the inclusion of participants who possessed a wealth of information [24].

Point 8: Line 133-134: How was the sample saturation determined in this study? Please be specific.

Response 8: Sample saturation was achieved when there were repeatable data during data collection and no additional new themes identified during analysis.

Changes to text:

Page 3, Line 128-132

The study's sample size was determined to be 40 individuals, as prior research has indicated that for cross-cultural or multi-sited inquiries, a sample size of 20-40 is necessary to attain data saturation [25]. The point of saturation was reached in the study when participants began providing identical responses and no further data was gathered from subsequent participants [26].

Point 9: Lines 134-136: Please explain what is meant by expressions such as “…will be met…” and “participants will then be approached…”?

Response 9: The expressions had been removed.

Point 10: Lines 138-139: Please provide a section to explain what was done to secure the confidentiality and privacy of participant information while they were interviewed in their homes.

Response 10: An ethical approval subsection is added.

Changes to text:

Page 4, Line 160-165

The confidentiality of participants was ensured by not collecting any personal information during the data collection procedure and keeping all data anonymous. The audio recording was transcribed and then deleted.

Point 11: Lines 138-139: Please be specific about the number of interviews conducted, indicating how “40” was determined. Again, how was the sample saturation determined in this study?

Response 11: The study's sample size and decision for data saturation were based on previous studies.

Changes to text:

Page 3, Line 128-132

The study's sample size was determined to be 40 individuals, as prior research has indicated that for cross-cultural or multi-sited inquiries, a sample size of 20-40 is necessary to attain data saturation [25]. The point of saturation was reached in the study when participants began providing identical responses and no further data was gathered from subsequent participants [26].

Point 12: Lines 139-142: Please describe the semi-structured interview guide development process. Was this interview guide pilot tested? Please describe this process in detail.

Response 12:  The semi-structured interview guide was not piloted.

Changes to text:

Page 3, Line 136-142

The study employed a semi-structured questionnaire that was designed based on the constructs of the Health Belief Model, including perceived susceptibility, perceived severity, perceived effectiveness, perceived barriers, cues to action, and self-efficacy. This approach was adopted to ensure uniformity in the data collection process among the participants. However, the interview inquiries were not constrained by the survey instrument. Additional inquiries were posed following the answer given by the respondents.

Point 13: Lines 139-142: How many questions were there in this interview guide initially?

Response 13: There were 10 open-ended questions in the interview guide initially based on the six  constructs of the Health Belief Model.

Point 14: Lines 139-142: Please provide a concise description of the “Health Belief Model” here and describe how it informed the interview guide development.

Response 14: Interview was done using a semi-structured questionnaire based on the constructs of the Health Belief Model (perceived susceptibility, perceived severity, perceived effectiveness, perceived barriers, cues to action, and self-efficacy). Questions were asked guided by the definition provided in Table 1. The questions were attached in the interview guide attached.

Point 15: Lines 139-142: Please describe if there were any debriefings after interviews were conducted?

Response 15: Debriefings was done after data collection weekly since principal investigator entered study site once a week. 

Changes to text:

Page 3, Line 144-145

Following each data collection, debriefings were conducted to refine and enhance the interviewing process [27].

Point 16: Lines 139-142: Please describe if there were any iterative refinement of the interview guide. Did the number of questions change as a result? Please be specific.

Response 16: The interview questions were not restricted by the questionnaire as majority of the marginalised population came from the lower socioeconomic background. The way the questions were asked could change depending on the participant’s comprehension and how they answered them.  

Changes to text:

Page 3, Line 140-141

The way the questions were asked could change depending on how the participants answered them.

Point 17: Lines 139-142: Please provide a copy of the interview guide in an appendix.

Response 17: Copy of interview guide is added in the appendix.

Point 18: Lines 142-145: It is better practice that the initials of the “principal investigator” is provided. Is the “principal investigator” and “primary researcher” the same individual? Please be precise.

Response 18: The term principal investigator and primary researcher was meant to be the same person. Thus, the usage of principal investigator has been standardised in this manuscript.

Point 19: Lines 147-156: This reviewer is very concerned that the analytical process was not conducted by at least 2 authors independently which brings into question the credibility of results.

Response 19: The sentences on the way data analysis was done in this study were edited for better comprehension.

Changes to text:

Page 4, Line 154-156

The coding and analysis procedures were executed independently by two researchers. The accuracy of the themes was evaluated by comparing the obtained results.

Point 20: Lines 147-156 and Table 1: Please describe the data analysis process clearly (for example, is it the use of “content analysis” that “employed parts of the Health Belief Model…to code data…” or is it the “inductive analysis of qualitative data” that “identified common themes…” as in here). Also, see the Title and lines 427-429 for similar concerns that this reviewer have given that the analysis process is not clearly described.

Response 20: The data analysis part was edited.

Changes to text:

Page 4, Line 146-157

The thematic analysis employed parts of the Health Belief Model as a theoretical lens to code data [28,29]. The HBM postulates that people are more or less motivated to take action based on their judgements of their own vulnerability, the severity of a problem, and the benefits and costs of addressing that problem. The likelihood of behaviour change is also affected by things like internal and external cues to act (like the media) and a person’s confidence in their own ability to act (self-efficacy) [30].

Emergent themes were identified from the analysis. The transcripts were inputted and analysed using the Atlas.ti 22 software. The coding and analysis procedures were executed independently by two researchers. The accuracy of the themes was evaluated by comparing the obtained results. Table 1 shows how each theory construct was used in this research.

Point 21: Lines 147-156: Again, who is “principal analyst”? Please provide initials.

Response 21: The principal analyst is removed.

Point 22: Lines 147-156: Again, which “member of the research team”? Please provide initials.

Response 22: The sentence member of research team was removed.

Point 23: Lines 147-156: What did that process of “check[ing] the coding afterwards” involve?

Response 23: The sentence checking the coding was removed.

Changes to text:

Page 4, Line 154-156

The coding and analysis procedures were executed independently by two researchers. The accuracy of the themes was evaluated by comparing the obtained results.

Point 24: Lines 154-155: How are “groups” similar or different from “themes”?

Response 24: The word “groups” were removed.

Point 25: Lines 155-156: How was it possible to “finally find recurring patterns or themes”?

Response 25: This sentence was removed and replaced with clearer sentence.

Changes to text:

Page 4, Line 154-156

The coding and analysis procedures were executed independently by two researchers. The accuracy of the themes was evaluated by comparing the obtained results.

Point 26: After line 156: Please add here to describe if there were ‘member checking’.

Response 26: There was no member checking but another member did the coding and result was compared. The sentence was edited.

Page 4, Line 154-156

The coding and analysis procedures were executed independently by two researchers. The accuracy of the themes was evaluated by comparing the obtained results.

Point 27: T Lines 159-164 & Table 2: “Nil”?

Response 27: Table 2 was summarised.

Changes to text:

Page 4-5, Line 170-181

Table 2 summarised the characteristics of the participants. All 40 participants in the study were women who identified as full-time housewives. The age range of the study participants was between 17 and 64 years. Among them, 13 individuals belonged to the young mother cohort, aged between 17 and 24 years, while 26 participants were aged between 25 and 44 years. Only one participant was aged 64 years. Among the entire cohort of participants, a subset of 13 individuals did not undergo any formal education, whereas another subset of 17 individuals received primary education at a minimum. Furthermore, a total of 10 individuals were provided with formal education up to the level of high school. Moreover, out of the 40 participants, four individuals opted not to obtain measles immunisation for their children, whereas 30 participants ensured that their children received the vaccine. Two parents had children with incomplete measles vaccination, while four parents had children who had not yet reached the age to receive measles immunisation.

Point 28: The rest of the sections of this manuscript: Given above points, this reviewer is unsure in terms of how to continue with the rest of the section of this manuscript other than to indicate that direct quotes should be translated to English (since the rest of manuscript is in English).

Response 28: The quotes were all translated to English.

Changes to text:

Page 5-9, Line 201-338

Point 29: Please avoid typos and ensure completeness, consistency and transparency in the manuscript: a) please make sure that all citations are referenced in a similar format in the body of the manuscript (see, for example, “Anna A. Minta….” in line 68, why is this citation in a different format?), b) please ensure all citations in the body of the manuscript are also reported in the Reference section (for example, see above for the missing one), c) please ensure completeness of citations and follow the same format throughout the Reference section (for example, why are some journal names abbreviated and others not, why some citations start with first name of the author and others with the last name, why are some citations in capital letters, what does “xxxx” mean…), d) please be transparent (which “report, line 68; which “region” line 69…), e) please write out all acronyms (such as, “MCV” in line 78 or “HBM” in line 119) the first time they are mentioned, f) please do not repeat writing out the acronyms later in the manuscript (see lines 166-170 for “HBM”), g)  please ensure consistency (see, for example, is it “majority…” or is it “only a few…” lines 317-319), and h) please proofread before submission (see, for example, “…with other literature in a According to a Steven et al. study…” lines 359-361)…

Response 29: References were double checked and manuscript sent for proofreading prior to journal submission.

Reviewer 3 Report

The paper is well written and appropriate within the context.

The sample size of 40 is rather small. What is the population size in context?

Authors may include a paragraph discussion in introduction comparing measles to other diseases/vaccination efforts...e.g., polio, etc. to provide a contrast about the contemporary status of vaccination.

Other scope & limits may be identified, e.g., sample size, quality of data, causality v. correlation, etc.

Author Response

Point 1: The paper is well written and appropriate within the context.

The sample size of 40 is rather small. What is the population size in context?

Response 1: The population size for both study sites was around 1000. Usage of sample size 40 and data saturation decision were based on previous research. 

Changes to text:

Page 3, Line 128-132

The study's sample size was determined to be 40 individuals, as prior research has indicated that for cross-cultural or multi-sited inquiries, a sample size of 20-40 is necessary to attain data saturation [25]. The point of saturation was reached in the study when participants began providing identical responses and no further data was gathered from subsequent participants [26].

Point 2: Authors may include a paragraph discussion in introduction comparing measles to other diseases/vaccination efforts...e.g., polio, etc. to provide a contrast about the contemporary status of vaccination.

Response 2: Sentence regarding this has been added.

Changes to text:

Page 2, Line 56-63

Community volunteering initiative also had shown success in assisting the polio supplementary immunisation activities in Sabah [4]. The benefits of volunteering action include empowerment, increased resilience and social connectedness [5], fostering a sense of neighbourhood trust and communal belonging [6], increasing members’ emotions of support, and boosting mental health [7].

               Based on the various advantages, community volunteering should be applied to other health initiatives, including the measles immunisation programme, where measles re-mains an endemic disease in Malaysia.

Point 3: Other scope & limits may be identified, e.g., sample size, quality of data, causality v. correlation, etc.

Response 3: Limitations were added.

Changes to text:

Page 13, Line 459-465

There were several limitations in this study. Firstly, was the involvement of only two study sites due to budget and time constraints. Secondly, there could be information bias since the participants were all women. Thirdly, having a bigger sample size with involvement of participants from other villages would allow further representation of the study findings towards the marginalised population. Finally, having a piloted semi-structured questionnaire would produced a better-quality study.

Round 2

Reviewer 2 Report

Thanks to authors for their revisions.

The following points require further attention. Old numbering is kept for convenience.

1. Old “Lines 1-5: Please ensure that there are no typos in the Title.”

Thanks for indicating that the manuscript was sent for editing. This reviewer thinks that it should be “…Beliefs and Perceptions…”

2. Old “Lines 14-37: Please ensure that there are no typos in the Abstract.”

Thanks for indicating that the manuscript was sent for editing. This reviewer thinks that, for example, it should be “…at the ages six, nine…”

3. Old “Lines 42-125: Please consider shortening the Introduction section making it concise. The last paragraph should clearly state the gap in the literature and how this study aims to fill that gap. Also, the last paragraph should not be used to introduce new information such as “behavioural theory” (which can be done later in the Methods section).”

Thanks for revisions but the Introduction section still has citations that are referenced in different formats. Please also see the last review point below on this.

4. Old “Line 51: “(MyVac, 2022)”?”

Thanks for adding a reference. Please ensure that the reference is complete.

8. Old “Line 133-134: How was the sample saturation determined in this study? Please be specific.”

Thanks for adding information here and reiterating later internally as a response to review point # 11 below that “study’s sample size and decision for data saturation were based on previous studies”. Thanks also including this issue in the limitations section. This reviewer thinks that, in qualitative research, data saturation is study-specific, is determined through the processes used during data collection and cannot be determined on a priori basis with information from previous studies. Could authors be specific in their explanations? Is it that the sample size was predetermined to be 40 based on previous studies (as indicated in new lines 124-126)? At the same time, is it the case that sample saturation for this study was achieved at that exact same number of 40, that is the predetermined sample size (as indicated in new lines 126-128)?

11. Old “Lines 138-139: Please be specific about the number of interviews conducted, indicating how “40” was determined. Again, how was the sample saturation determined in this study?”

Thanks for revisions. Please see above.

12. Old “Lines 139-142: Please describe the semi-structured interview guide development process. Was this interview guide pilot tested? Please describe this process in detail.”

Thanks for indicating internally that the semi-structured interview guide was not pilot tested and thanks also for incorporating that later in the limitation section. Please incorporate this information into the Methods section and please provide a rationale. Also “survey instrument”?

13. Old “Lines 139-142: How many questions were there in this interview guide initially?”

Thanks for indicating internally that there were “10 questions”. Please incorporate that there were “10 questions” into the body of the manuscript where it is indicated that interviews took “…20 to 30 minutes” (new lines 136-141) so that the readership will have a perspective on average time allotted to each of those questions and, as indicated, to “…additional inquiries…”

16. Old “Lines 139-142: Please describe if there were any iterative refinement of the interview guide. Did the number of questions change as a result? Please be specific.”

Thanks for revisions. This review point is referring to the “refinement of the interview guide” itself as one of the processes used for this qualitative study during data collection.

17. Old “Lines 139-142: Please provide a copy of the interview guide in an appendix.”

Thanks for incorporating a supplemental file in this round of review. However, this reviewer could not find the copy of the interview guide itself in that supplemental file. The supplemental file provided just contains the front matter for the interview guide.

18. Old “Lines 142-145: It is better practice that the initials of “principal investigator” is provided. Is the “principal investigator” and “primary researcher” the same individual? Please be precise.”

Thanks, please provide initials.

19. Old “Lines 147-156: This reviewer is very concerned that the analytical process was not conducted by at least 2 authors independently which brings into question the credibility of results.”

Thanks. Please provide the initials of those 2 “researchers”. Again, please also be specific if one of the “researchers” identified here is also the “principle investigator” identified in new line 131, for example. Please make this point clear for the readership and, as before, please always include initials.

20. Old “Lines 147-156 and Table 1: Please describe the data analysis process clearly (for example, is it the use of “content analysis” that “employed parts of the Health Belief Model…to code data…” or is it the “inductive analysis of qualitative data” that “identified common themes…” as in here). Also, see the Title and lines 427-429 for similar concerns that this reviewer have given that the analysis process is not clearly described.”

Thanks for revisions. As before, initials are needed for those 2 “researchers”. Please see above.

23. Old “Lines 147-156: What did that process of “check[ing] the coding afterwards” involve?”

Thanks for revisions and deletions. Please describe what happened if “comparing the obtained results” resulted in conflicting “themes”?

25. Old “Lines 155-156: How was it possible to “finally find recurring patterns or themes”?”

Thanks for revisions and deletions. It is still unclear to this reviewer how the process (as currently described in this revised manuscript) evolved to determine the “accuracy of the themes” especially given the uncertainties surrounding the saturation process and the lack of evidence indicating that iterative processes were used during interview guide development and data analysis…

26. Old “After line 156: Please add here to describe if there were ‘member checking’.”

Thanks for revisions. This review point is referring to checking with the participants.

28. Old “The rest of the sections of this manuscript: Given above points, this reviewer is unsure in terms of how to continue with the rest of the section of this manuscript other than to indicate that direct quotes should be translated into English (since the rest of manuscript is in English).”

Thanks for translations. However, methodological concerns that this reviewer previously had remain. Please see above.

29. Old “Please avoid typos and ensure completeness, consistency and transparency in the manuscript: a) please make sure that all citations are referenced in a similar format in the body of the manuscript (see, for example, “Anna A. Minta….” in line 68, why is this citation in a different format?), b) please ensure all citations in the body of the manuscript are also reported in the Reference section (for example, see above for the missing one), c) please ensure completeness of citations and follow the same format throughout the Reference section (for example, why are some journal names abbreviated and others not, why some citations start with first name of the author and others with last name, why are some citations in capital letters, what does “xxxx” mean…), d) please be transparent (which “report, line 68; which “region” line 69…), e) please write out all acronyms (such as, “MCV” in line 78 or “HBM” in line 119) the first time they are mentioned, f) please do not repeat writing out the acronyms later in the manuscript (see lines 166-170 for “HBM”), g)  please ensure consistency (see, for example, is it “majority…” or is it “only a few…” lines 317-319), and h) please proofread before submission (see, for example, “…with other literature in a According to a Steven et al. study…” lines 359-361)…”

Thanks for revisions, however, all of the sections of the manuscript need to follow these (see, for example, above point #3 as a reference for section (a) here not being completed; or see, for example, new lines 143-146 or new lines 182-185, for section (e) here).

Author Response

Response to Reviewer 2 Comments

Point 1: Comments and Suggestions for Authors. Thanks to authors for their revisions. The following points require further attention. Old numbering is kept for convenience.
Old "Lines 1-5: Please ensure that there are no typos in the Title."

Thanks for indicating that the manuscript was sent for editing. This reviewer thinks that it should be "…Beliefs and Perceptions…"

Thank you to the reviewer for all the insightful remarks.

Response 1:  The title is corrected to as what the reviewer suggested. The manuscript is sent again for proofreading which this time is done by native English speaker. Certificate is attached.

Changes to text

Page 1, Line 2-5

A Behavioural Theory-based Qualitative Study of the Beliefs and Perceptions of Marginalised Populations towards Community Volunteering to Increase Measles Immunisation Coverage in Sabah, Malaysia.

Point 2: Old "Lines 14-37: Please ensure that there are no typos in the Abstract."
Thanks for indicating that the manuscript was sent for editing. This reviewer thinks that, for example, it should be "…at the ages six, nine…"

Response 2:  The sentence is corrected as the reviewer mentioned. The abstract was proofread again. Hence, the grammar was corrected.

Changes to text

Page 1, Line 17- 18

In the state of Sabah in Malaysia, a complete course of measles immunisation for infants involves vaccinations at the ages of six, nine, and twelve months.

Point 3: Old "Lines 42-125: Please consider shortening the Introduction section making it concise. The last paragraph should clearly state the gap in the literature and how this study aims to fill that gap. Also, the last paragraph should not be used to introduce new information such as "behavioural theory" (which can be done later in the Methods section)."

Thanks for revisions but the Introduction section still has citations that are referenced in different formats. Please also see the last review point below on this.

Response 3:  Thank you for this review. We noted the weakness for this current manuscript where the citation was in different format as seen from the point 4 below. Hence, we double check again each of the citations following the Reference List and Citations Style Guide for MDPI Journals. Corrections had been made.

Point 4: Old "Line 51: "(MyVac, 2022)"?"

Thanks for adding a reference. Please ensure that the reference is complete.

Response 4:  Thank you for the comment. We take note of the incompleteness and had double check and make improvement towards all the references and citations to ensure they follow MDPI format.

Changes to Text

Page 15, Line 536

  1. Minta, A.A.; Ferrari, M.; Antoni, S.; Portnoy, A.; Sbarra, A.; Lambert, B.; Haurisky, S.; Hatcher, C.; Nedelec, Y.; Datta, D.; et al. Progress Toward Regional Measles Elimination — Worldwide, 2000 – 2021. MMWR Morb Mortal Wkly Rep 2021, 2022, 71, doi:10.15585/mmwr.mm7147a1.

Point 5: 8. Old "Line 133-134: How was the sample saturation determined in this study? Please be specific."

Thanks for adding information here and reiterating later internally as a response to review point # 11 below that "study's sample size and decision for data saturation were based on previous studies". Thanks also including this issue in the limitations section. This reviewer thinks that, in qualitative research, data saturation is study-specific, is determined through the processes used during data collection and cannot be determined on a priori basis with information from previous studies. Could authors be specific in their explanations? Is it that the sample size was predetermined to be 40 based on previous studies (as indicated in new lines 124-126)? At the same time, is it the case that sample saturation for this study was achieved at that exact same number of 40, that is the predetermined sample size (as indicated in new lines 126-128)?

Response 5:  Thank you for the enquiry. The data saturation for this research was reached at the sample number 40 when the respondents provide similar response and no new added coding were added. Thus, this is supported by the previous studies referred to which mentioned that a sample size of 20-40 is necessary for a cross-cultural and multi-sited inquiry. We take note that a better explanation is needed. Hence, we rephrase the sentence to avoid misunderstanding.

Changes to text:

Page 3, Line 115-120

Data saturation; the point at which respondents begin providing similar responses; occurred once 40 respondents had been interviewed. As such, no additional data was collected from the subsequent respondents [24]. Extant studies indicate that a cross-cultural or multi-sited study requires a sample size of 20 to 40 respondents to achieve data saturation [25]. Furthermore, a larger sample size may not necessarily yield more valuable data in such studies. Therefore, the sample size of this present study (40) was optimal.

Point 6: 11. Old "Lines 138-139: Please be specific about the number of interviews conducted, indicating how "40" was determined. Again, how was the sample saturation determined in this study?"

Thanks for revisions. Please see above.

Response 6:  Forgive us for the way the sentence cause misunderstanding towards the reviewer. As such, the sentence had been rephrased as point 5 above.

Page 3, Line 115-120

Data saturation; the point at which respondents begin providing similar responses; occurred once 40 respondents had been interviewed. As such, no additional data was collected from the subsequent respondents [24]. Extant studies indicate that a cross-cultural or multi-sited study requires a sample size of 20 to 40 respondents to achieve data saturation [25]. Furthermore, a larger sample size may not necessarily yield more valuable data in such studies. Therefore, the sample size of this present study (40) was optimal.

Point 7: 12. Old "Lines 139-142: Please describe the semi-structured interview guide development process. Was this interview guide pilot tested? Please describe this process in detail."

Thanks for indicating internally that the semi-structured interview guide was not pilot tested and thanks also for incorporating that later in the limitation section. Please incorporate this information into the Methods section and please provide a rationale. Also "survey instrument"?

Response 7:  We take note of the reviewer's suggestion and add a section for survey instruments into the manuscript.

Changes to text:

Page 3, Line 122-135

2.3. Survey Instrument

               A dual-language semi-structured questionnaire was designed in both English and Malay on the six components of the Health Belief Model (HBM); namely, perceived susceptibility, perceived severity, perceived effectiveness, perceived barriers, cues to action, and self-efficacy; to standardise that the process that was used to collect data from the respondents. Nevertheless, the survey instrument did not affect the interview section of this present study as the answers that the respondents provided were often followed by additional questions from the interviewers. The HBM-based questionnaire was divided into two sections. The first section collected respondent demographics; such as age, gender, occupation, education level, and the immunisation history of their children while the second examined their beliefs and perceptions of measles immunisation and community-based volunteering. The 10 questions in the second section were open-ended to enable the respondents to share their ideas and opinions on the matter freely. The guidelines of the interview are included in the Supplementary File.

Point 8: 13. Old "Lines 139-142: How many questions were there in this interview guide initially?"

Thanks for indicating internally that there were "10 questions". Please incorporate that there were "10 questions" into the body of the manuscript where it is indicated that interviews took "…20 to 30 minutes" (new lines 136-141) so that the readership will have a perspective on average time allotted to each of those questions and, as indicated, to "…additional inquiries…"

Response 8:  We noted the suggestions and had edited the study procedures section so that additional inquiries were addressed.

Changes to text:

Page 3-4, Line 137-151

2.4. Study Procedures

This present study was conducted over a nine-month period; from April to December 2022. The required data was collected by conducting in-depth one-on-one interviews with the respondents at their residences. The study's principal investigator, HS, visited the research site every week and used a semi-structured questionnaire as a guide to conduct the interviews. The mean duration of each interview was approximately 20 to 30 minutes. The audio of every interview was recorded then transcribed verbatim on a weekly basis. Debriefings were held every week after each phase of data collection to optimise and improve the research method [26]. The interview transcripts and codes were thorough reviewed by all members of the research team. Disagreements were discussed and resolved together. As all the respondents provided clear answers, clarification was not required. Throughout the interview period, the number of guided questions remained consistent and none of the questions were deemed unnecessary as all respondents demonstrated a comprehensive understanding of the questions posed. This was because the questions in the Malay language questionnaire were revised to better align with the local dialect.

Point 9: 16. Old "Lines 139-142: Please describe if there were any iterative refinement of the interview guide. Did the number of questions change as a result? Please be specific."

Thanks for revisions. This review point is referring to the "refinement of the interview guide" itself as one of the processes used for this qualitative study during data collection.

Response 9:  Thank you for the further clarification given regarding the comments. We therefore explained further on the iterative refinement of the interview guide in the study procedure section.

Changes to text:

Page 3-4, Line 147-150

Throughout the interview period, the number of guided questions remained consistent and none of the questions were deemed unnecessary as all respondents demonstrated a comprehensive understanding of the questions posed.

Point 10: 17. Old "Lines 139-142: Please provide a copy of the interview guide in an appendix."

Thanks for incorporating a supplemental file in this round of review. However, this reviewer could not find the copy of the interview guide itself in that supplemental file. The supplemental file provided just contains the front matter for the interview guide.

Response 10:  Forgive us for not attaching the supplemental file properly. We hoped the attached supplemental file this time is complete.

Point 11: 18. Old "Lines 142-145: It is better practice that the initials of "principal investigator" is provided. Is the "principal investigator" and "primary researcher" the same individual? Please be precise."

Thanks, please provide initials.

Response 11:  Initials were inserted.

Changes to text:

Page 4, Line 160-162

Themes were identified from the analysis. ATLAS.ti 22 was used to qualitatively assess the interview transcripts while two researchers, HS and MA, coded and analysed them independently.

Point 12:19. Old "Lines 147-156: This reviewer is very concerned that the analytical process was not conducted by at least 2 authors independently which brings into question the credibility of results."

Thanks. Please provide the initials of those 2 "researchers". Again, please also be specific if one of the "researchers" identified here is also the "principle investigator" identified in new line 131, for example. Please make this point clear for the readership and, as before, please always include initials.

Response 12: Thank you for the information. We double checked the manuscript again to ensure initials were attached at the necessary parts.

Changes to text:

Page  3,Line 140-142

The study's principal investigator, HS, visited the research site every week and used a semi-structured questionnaire as a guide to conduct the interviews.

Point 13: 20. Old "Lines 147-156 and Table 1: Please describe the data analysis process clearly (for example, is it the use of "content analysis" that "employed parts of the Health Belief Model…to code data…" or is it the "inductive analysis of qualitative data" that "identified common themes…" as in here). Also, see the Title and lines 427-429 for similar concerns that this reviewer have given that the analysis process is not clearly described."

Thanks for revisions. As before, initials are needed for those 2 "researchers". Please see above.

Response 13:  We had inserted the two initials as needed.

Changes to text:

Page 4, Line 160-162

Emergent themes were identified from the analysis. The transcripts were inputted and analysed using the Atlas.ti 22 software. The coding and analysis procedures were executed independently by two researchers, HS and MA.

Point 14:23. Old "Lines 147-156: What did that process of "check[ing] the coding afterwards" involve?"

Thanks for revisions and deletions. Please describe what happened if "comparing the obtained results" resulted in conflicting "themes"?

Response 14:  We had added the sentences in regards to what happened when there is conflicting themes.

Changes to text:

Page 4, Line 162-164

The accuracy of the themes was determined by comparing the obtained results. Conflicting themes were discussed amongst themselves as well as with every member of the research team to arrive at a final decision.

Point 15: 25. Old "Lines 155-156: How was it possible to "finally find recurring patterns or themes"?"

Thanks for revisions and deletions. It is still unclear to this reviewer how the process (as currently described in this revised manuscript) evolved to determine the "accuracy of the themes" especially given the uncertainties surrounding the saturation process and the lack of evidence indicating that iterative processes were used during interview guide development and data analysis…

Response 15:  We had edited the study procedures to ensure further explanation regarding how the research is clear. The saturation process is explained in Point 6.

Changes to text:

Page 3-4, Line 137-151

2.4. Study Procedures

This present study was conducted over a nine-month period; from April to December 2022. The required data was collected by conducting in-depth one-on-one interviews with the respondents at their residences. The study's principal investigator, HS, visited the research site every week and used a semi-structured questionnaire as a guide to conduct the interviews. The mean duration of each interview was approximately 20 to 30 minutes. The audio of every interview was recorded then transcribed verbatim on a weekly basis. Debriefings were held every week after each phase of data collection to optimise and improve the research method [26]. The interview transcripts and codes were thorough reviewed by all members of the research team. Disagreements were discussed and resolved together. As all the respondents provided clear answers, clarification was not required. Throughout the interview period, the number of guided questions remained consistent and none of the questions were deemed unnecessary as all respondents demonstrated a comprehensive understanding of the questions posed. This was because the questions in the Malay language questionnaire were revised to better align with the local dialect.

Point 16: 26. Old "After line 156: Please add here to describe if there were 'member checking'."

Thanks for revisions. This review point is referring to checking with the participants.

Response 16:  We noted that we misunderstood the previous review given. Hence, we add a sentence to answer the review point on checking with the participants.

Changes to text:

Page 3, Line 147

As all the respondents provided clear answers, clarification was not required.

Point 17: 28. Old "The rest of the sections of this manuscript: Given above points, this reviewer is unsure in terms of how to continue with the rest of the section of this manuscript other than to indicate that direct quotes should be translated into English (since the rest of manuscript is in English)."

Thanks for translations. However, methodological concerns that this reviewer previously had remain. Please see above.

Response 17:  We sincerely hope the adjustment that had been made above addressed the reviewer’s methodological concern.

Point 18: 29. Old "Please avoid typos and ensure completeness, consistency and transparency in the manuscript:

  1.  please make sure that all citations are referenced in a similar format in the body of the manuscript (see, for example, "Anna A. Minta…." in line 68, why is this citation in a different format?),
  2.  please ensure all citations in the body of the manuscript are also reported in the Reference section (for example, see above for the missing one),
  3. please ensure completeness of citations and follow the same format throughout the Reference section (for example, why are some journal names abbreviated and others not, why some citations start with first name of the author and others with last name, why are some citations in capital letters, what does "xxxx" mean…),
  4. please be transparent (which "report, line 68; which "region" line 69…),
  5. please write out all acronyms (such as, "MCV" in line 78 or "HBM" in line 119) the first time they are mentioned,
  6. please do not repeat writing out the acronyms later in the manuscript (see lines 166-170 for "HBM"),
  7.  please ensure consistency (see, for example, is it "majority…" or is it "only a few…" lines 317-319), and
  8. h) please proofread before submission (see, for example, "…with other literature in a According to a Steven et al. study…" lines 359-361)…"

    Thanks for revisions, however, all of the sections of the manuscript need to follow these (see, for example, above point #3 as a reference for section (a) here not being completed; or see, for example, new lines 143-146 or new lines 182-185, for section (e) here).

Response 18: -

Response 18 a) Thank you for informing. Thus, we double checked every single citation and adjusted each one to follow the follow MDPI format.

Response 18 b) We realised the mistake and had ensured all the citations were reported in the reference section.

Response 18 c) We double checked all the articles to ensure that the last name of the authors come first according to MDPI format, ensure that the journal names were not abbreviated, and ensure other details follows the format

Response 18 d) Regarding ‘report’ and ‘region’, we had change the sentences which in the end did not need to use these words.

Response 18 e) We noted the change needed and double checked the acronyms

Response 18 f) We noted the mistake and double checked the acronyms to make sure it is not repeated.

Response 18 g) We noted and made some editing to ensure consistency on the use of words when classifying the number of quotes.

Response 18 h) We noted the need for another proofreading done by native speaker.

Reviewer 3 Report

Authors have addressed the concerns.

Author Response

Response to Reviewer 3 Comments

Point 1:  English language and style are fine/minor spell check required

Response 1: Thank you for the review. We noted the weakness of this manuscript, thus we resent the manuscript for another proofreading where this time it was done by English native-speaker.

Point 2: Is the research design appropriate? Can be improved

Response 2: We noted the research design could be improved especially since the methodology part of this research may have not been explained clearly. So, we had edited the methodology part extensively. We added survey instrument section and added more explanation in the data procedures.

Change to text

Page 3-4, Line 104-165

2.2. Sample selection and recruitment

The respondents of this present study were Malaysian citizens from marginalised communities living in urban slums and squatter areas as well as both legal and illegal migrants. Purposeful sampling was used to recruit the intended respondents; namely, parents or caregivers with knowledge of childhood immunisation. Purposeful sampling was used as it guaranteed the inclusion of respondents who possess a wealth of information [23]. Data saturation; the point at which respondents begin providing similar responses; occurred once 40 respondents had been interviewed. As such, no additional data was collected from the subsequent respondents [24]. Extant studies indicate that a cross-cultural or multi-sited study requires a sample size of 20 to 40 respondents to achieve data saturation [25]. Furthermore, a larger sample size may not necessarily yield more valuable data in such studies. Therefore, the sample size of this present study (40) was optimal.

2.3. Survey Instrument

               A dual-language semi-structured questionnaire was designed in both English and Malay on the six components of the Health Belief Model (HBM); namely, perceived susceptibility, perceived severity, perceived effectiveness, perceived barriers, cues to action, and self-efficacy; to standardise that the process that was used to collect data from the respondents. Nevertheless, the survey instrument did not affect the interview section of this present study as the answers that the respondents provided were often followed by additional questions from the interviewers. The HBM-based questionnaire was divided into two sections. The first section collected respondent demographics; such as age, gender, occupation, education level, and the immunisation history of their children while the second examined their beliefs and perceptions of measles immunisation and community-based volunteering. The 10 questions in the second section were open-ended to enable the respondents to share their ideas and opinions on the matter freely. The guidelines of the interview are included in the Supplementary File.

2.4. Study Procedures

This present study was conducted over a nine-month period; from April to December 2022. The required data was collected by conducting in-depth one-on-one interviews with the respondents at their residences. The study's principal investigator, HS, visited the research site every week and used a semi-structured questionnaire as a guide to conduct the interviews. The mean duration of each interview was approximately 20 to 30 minutes. The audio of every interview was recorded then transcribed verbatim on a weekly basis. Debriefings were held every week after each phase of data collection to optimise and improve the research method [26]. The interview transcripts and codes were thorough reviewed by all members of the research team. Disagreements were discussed and resolved together. As all the respondents provided clear answers, clarification was not required. Throughout the interview period, the number of guided questions remained consistent and none of the questions were deemed unnecessary as all respondents demonstrated a comprehensive understanding of the questions posed. This was because the questions in the Malay language questionnaire were revised to better align with the local dialect.

2.5. Data Analysis

A thematic analysis was conducted using the components of the HBM to theoretically code the collected data [27,28]. According to the HBM, individuals are more or less motivated to perform an action based on their perceptions of their own vulnerability to a problem, the severity of the problem, and the benefits and cost of addressing the problem. Apart from that, internal and external cues to act; from the media, for example; as well as an individual's confidence in their own ability to act or self-efficacy affect the likelihood of changing a behaviour [29].

Themes were identified from the analysis. ATLAS.ti 22 was used to qualitatively assess the interview transcripts while two researchers, HS and MA, coded and analysed them independently. The accuracy of the themes was determined by comparing the obtained results. Conflicting themes were discussed amongst themselves as well as with every member of the research team to arrive at a final decision. Table 1 depicts how each theoretical construct was used in the present study.